# The Presence and Consequences of Abortion Aversion in Scientific Research Related to Alcohol Use during Pregnancy

**DOI:** 10.3390/ijerph16162888

**Published:** 2019-08-13

**Authors:** Sarah C.M. Roberts

**Affiliations:** Advancing New Standards in Reproductive Health (ANSIRH), Department of Obstetrics, Gynecology, and Reproductive Sciences, University of California, San Francisco, Oakland, CA 94143, USA; sarah.roberts@ucsf.edu; Tel.: +1-510-986-8962

**Keywords:** alcohol, pregnancy, abortion, policy

## Abstract

Recent research has found that most U.S. state policies related to alcohol use during pregnancy adversely impact health. Other studies indicate that state policymaking around substance use in pregnancy—especially in the U.S.—appears to be influenced by an anti-abortion agenda rather than by public health motivations. This commentary explores the ways that scientists’ aversion to abortion appear to influence science and thus policymaking around alcohol and pregnancy. The three main ways abortion aversion shows up in the literature related to alcohol use during pregnancy include: (1) a shift from the recommendation of abortion for “severely chronic alcoholic women” to the non-acknowledgment of abortion as an outcome of an alcohol-exposed pregnancy; (2) the concern that recommendations of abstinence from alcohol use during pregnancy lead to terminations of otherwise wanted pregnancies; and (3) the presumption of abortion as a negative pregnancy outcome. Thus, abortion aversion appears to influence the science related to alcohol use during pregnancy, and thus policymaking—to the detriment of developing and adopting policies that reduce the harms from alcohol during pregnancy.

## 1. Background

Alcohol is a known teratogen that causes fetal alcohol syndrome (FAS) and a range of other harms to fetuses [1,2,3,4,5]. As many as 14.6 per 10,000 people worldwide may have FAS [6]. Alcohol use during pregnancy is common, with about 10% of pregnant women worldwide reporting any alcohol use [7]. In the U.S., 15% of pregnant women report any alcohol use and approximately 3% report binge drinking in the past month [8]. Rates of alcohol use during pregnancy are higher in some regions, such as Europe, and lower in others, such as the eastern Mediterranean, and Southeast Asia [7]. In the U.S., while there have been some minor fluctuations, rates of alcohol use during pregnancy have remained steady since the 1990s [9,10,11,12,13]. This means that alcohol use during pregnancy has remained common for decades in the U.S., despite considerable governmental and clinical attention in the U.S. and other countries.

To address drinking in pregnancy, many U.S. states have passed laws related to alcohol use in pregnancy. In the U.S., most states have at least one policy focusing on alcohol use during pregnancy (alcohol/pregnancy policies), and the number of alcohol/pregnancy policies has increased dramatically over the past 40 years [14]. However, until recently, there has been little quantitative research about the impacts of these policies. A recent comprehensive legal epidemiology [15] study found that, in the U.S., a few state alcohol/pregnancy policies may be associated with less self-reported drinking during pregnancy [16]. This legal epidemiology study also found that most state alcohol/pregnancy policies, at best, have no relationship to birth outcomes (such as low birth weight and preterm birth) or prenatal care use [17]. At worst, using methods that allow for causal inference, this study found that multiple state alcohol/pregnancy policies lead to increases in low birth weight and preterm birth, and decreases in prenatal care use [17]—leading to thousands of babies born with low birth weight, or preterm each year, due to the policies [18].

A recent study of state legislators in three U.S. states found that while state legislators are aware that drinking during pregnancy is harmful, they also believe (inaccurately) that “nobody does that any more” due to public health efforts [19]. Despite these beliefs and the lack of research evidence about the effectiveness of alcohol/pregnancy policies, these states have multiple alcohol/pregnancy policies and have recently adopted new alcohol/pregnancy policies [14]. This suggests a strong disconnect between research evidence and policy related to alcohol use during pregnancy.

One explanation for this disconnect is that U.S. policymaking related to alcohol as well as drug use during pregnancy has been influenced by an anti-abortion political agenda, having to do with treating the fetus as a separate person from the pregnant woman [20,21] rather than being driven by public health efforts designed to reduce harms from alcohol use during pregnancy [14]. For context, over the same 40-year time period that the number of state alcohol/pregnancy policies increased, there has also been a dramatic increase in the number of state-level policies restricting abortion in the U.S. [22]. While other countries where abortion is legal also have some restrictive abortion policies and limited availability of abortion services [23], politics related to abortion in the U.S. have been more ever present than in many other countries [24]. In the U.S., anti-abortion activism and politics emerged as a strong political force in the early 1980s [24,25], and has remained a major issue in U.S. politics since then [26]. However, the terms of the U.S. political debate are narrowly framed. While politicians opposed to abortion directly state their opposition, abortion rights supporters do not typically frame abortion or even the availability of abortion services as positive. Instead, they largely frame abortion as something to be avoided as much as possible, using the slogan of “safe, legal, and rare” to describe a vision for abortion [27]. Essentially, even abortion rights supporters in the U.S. use language that indicates an aversion to abortion.

This same aversion to abortion that is present in U.S. politics in general is also present in the scientific research related to alcohol use during pregnancy. The Merriam–Webster dictionary defines aversion as a feeling of repugnance toward something, with a desire to avoid or turn away from it, and as a tendency to extinguish a behavior or to avoid a thing or situation [28]. In this commentary, using this definition of aversion as a guide, I trace three main ways aversion to abortion appears to influence scientific questions asked and thus perhaps policy options scholars and other health professionals imagine related to alcohol (and drug) use during pregnancy. The three main ways that abortion aversion shows up in this literature include: “Shift from recommendation of abortion to non-acknowledgement of abortion as an outcome of alcohol or drug-exposed pregnancy,” “Concern that abstinence messages lead to terminations of otherwise wanted pregnancies,” and “Presumption of abortion as negative pregnancy outcome.” This commentary describes each aspect of abortion aversion and argues that abortion aversion appears to influence science related to alcohol use during pregnancy, and thus policymaking—to the detriment of developing and adopting policies that reduce harms from alcohol during pregnancy.

## 2. Shift from Recommendation of Abortion to Non-Acknowledgement of Abortion as an Outcome of Alcohol-Exposed Pregnancy

### 2.1. Early Recommendation of Abortion

FAS had been identified multiple times in past centuries and earlier in the 1900s, before the diagnosis publicly and medically “stuck” in the 1970s [29,30]. Historian Janet Golden and sociologist Elizabeth Armstrong argued that experiences with thalidomide in the 1950s that allowed the imagining of the fetus as a separate being that could be harmed by a pregnant woman’s behavior, contributed to the diagnosis sticking in the 1970s [29,31]. Golden also argued that the new availability of legal abortion services in 1973 that resulted from the U.S. Supreme Court Decision (Roe v. Wade) that legalized abortion throughout the U.S. was a key component of making the diagnosis of FAS stick during the 1970s. Essentially, legal abortion was a medical solution to the problem of FAS [29,31].

In fact, multiple peer-reviewed manuscripts published in the 1970s recommended or suggested abortion as a possible “solution” to (heavier) alcohol use during pregnancy, and to FAS [32,33,34]. This recommendation of abortion as a “solution” was made in one of the first manuscripts about FAS published by the medical researchers who “discovered” FAS in the U.S. in the 1970s [32]. In the manuscript, Jones and Smith concluded, “The frequency (43%) of adverse outcome of pregnancy for chronic alcoholic women suggests that serious consideration be given to early termination of pregnancy in severely chronic alcoholic women” (p. 1).

### 2.2. Disappearance of Abortion Recommendation

This focus on abortion as a “solution” to FAS essentially disappeared from the literature in the 1980s. There are a few possible explanations. First, Armstrong and Abel [35] argued that this abortion recommendation, which they characterized as “extreme,” was one aspect of the “moral panic” that emerged related to alcohol use during pregnancy in the 1970s and 1980s. They suggested that the abortion recommendation disappeared due to newer studies finding that FAS was a rarer outcome of alcohol use during pregnancy than in earlier studies, and thus that such an “extreme” solution was no longer necessary. The idea of abortion as an “extreme” solution can also be viewed as an element of abortion aversion—i.e., that abortion is to be avoided if at all possible.

Second, during the 1980s and 1990s, the War on Drugs and the emergence of anti-abortion politics in the U.S. shifted visual images and framing in broadcast news stories about women who drink during pregnancy [20]. Specifically, the images and framing shifted from portrayals of how government warnings about drinking during pregnancy might benefit primarily white, middle class women, and their children, to images of women who drink during pregnancy as women of color, who are deviant and are causing irreversible damage to their fetuses. With this shift to the idea of maternal substance use as causing fetal harm, came larger societal efforts to control pregnant women’s behavior more broadly—including both substance use during pregnancy and abortion [20].

Third, scientists in the 1980s and 1990s could have been influenced by a similar abortion aversion as that in U.S. politics. It is also possible that scientists, even if they did not share this aversion, were loath to stir controversy by directly addressing abortion in their work. They may also have been influenced by prohibitions in the U.S. on researching abortion provision using federal dollars [36].

### 2.3. Non-Acknowledgment of Abortion as a Possible Outcome

There are multiple instances in scientific literature where abortion is not acknowledged as a possible outcome of an alcohol-exposed pregnancy. For example, in the late 1990s, U.S. Centers for Disease Control and Prevention (CDC, the U.S. federal health agency) scientists started a body of research and interventions into what are now referred to as alcohol-exposed pregnancies, focusing on (often unintended) pregnancies in which women drank prior to discovering they were pregnant [37]. That some women who drink prior to discovering their pregnancy might have abortions (instead of giving birth) is absent from key highly cited studies—these studies created the idea of and set the research, policy, and intervention agendas for alcohol-exposed pregnancies (e.g., [37]).

A recent case illustrates how non-acknowledgement of abortion as a possible outcome influences the science about public health impacts of alcohol-exposed pregnancies. In 2016, scientists at the U.S. CDC used data about past month sexual behavior, contraception use, and alcohol use from a national survey to estimate the number of U.S. women at risk for an alcohol-exposed pregnancy [38]. The CDC produced multiple publicity materials to go along with their scientific report of this estimate [39,40]. The CDC publicity materials appeared to tell women of reproductive age that they should not drink unless they were using contraception [41]. There was a considerable pushback and, in fact, public ridicule and arguments to dismiss the entire premise of alcohol-exposed pregnancies [42,43,44]. Rather than have a public conversation about the harms related to alcohol use during pregnancy, communicate the importance of limiting drinking during pregnancy, and focus on solutions that might reduce harmful drinking [44], the conversation was mocking and dismissive of the CDC and their authority regarding harms from alcohol use during pregnancy. Importantly, abortion aversion appears present in this (seemingly counterproductive) effort. In their estimate, the CDC scientists used non-evidence-based assumptions related to abortion and abortion politics to draw inferences from the data available in the survey. First, they assumed pregnancies occur immediately after unprotected sex, i.e., prior to implantation. According to the American College of Obstetricians and Gynecologists and to U.S. government regulations, pregnancy occurs only after implantation [45,46], which takes place multiple days after fertilization [46]. Arguing that a pregnancy exists from the moment of fertilization is core to anti-abortion organizing in the U.S. [47]. The CDC authors also failed to acknowledge that not all alcohol-exposed pregnancies result in births, that some end in miscarriages and abortions. Along with these and other assumptions, their failure to acknowledge abortion as a possible outcome dramatically inflated the estimate of alcohol exposed pregnancies [48]. These inflated estimates are an element of and can contribute to moral panic related to alcohol use during pregnancy [35], and possibly influence the adoption of more harmful and stigmatizing policies.

As a second example, this non-acknowledgement of abortion as a possible outcome appears in clinical practice literature. Another highly cited study related to alcohol and pregnancy that was published in 2000, around the same time as the initial alcohol-exposed pregnancy article, and also by U.S. CDC scientists, explored what obstetricians and gynecologists need regarding their patients’ alcohol use during pregnancy [49]. In this manuscript, there is also no mention of abortion. The lack of acknowledgement of abortion as a possible outcome is especially surprising in this manuscript, as more than 80% of the obstetricians and gynecologists surveyed said that they wanted information on thresholds of alcohol consumption that cause reproductive harm [49]. Thresholds that cause reproductive harm (as well as information which may modify the risk of harm, such as nutrition and socioeconomic status [50,51,52,53]) are highly relevant to whether a pregnant woman who consumed alcohol during pregnancy might want to consider abortion. To spell this out, providing information about differences in the rates of harms at different levels of consumption (or under different conditions) might provide what appears to be justification for counseling on, or the suggestion of, abortion as an option for women drinking at certain levels or under certain other conditions during pregnancy. The failure to assess provider practices or information needs related to abortion in what appears to be an agenda-setting study essentially precludes development of a research agenda that might include this abortion-related topic.

Certainly, factors other than abortion aversion may have influenced the disappearance of abortion as a possible outcome for an alcohol-exposed pregnancy or as a “solution” for FAS. In relation to FAS and more so in relation to drug use during pregnancy, there is a deeply problematic legacy of efforts to use state and other forms of power to coerce and control the reproduction of women who use alcohol and drugs [54,55,56]. The one article that focuses on abortion as a possible outcome of alcohol and drug-exposed pregnancies is a case study of two psychiatric patients with substance use disorders who want to terminate their pregnancies; the authors of this case study carefully explain and detail the ethics process they went through to allow these women to obtain their wanted abortions [57]. There is no question that awareness of the potential for coercion, and engaging in intentional efforts to avoid contributing toward any justification for coercion, is warranted. However, such intense and seemingly overpowering concern about not contributing to coercion may also contribute to the non-acknowledgement of abortion as a possible outcome, and thus limit research questions asked and policies imagined.

## 3. Concern that Abstinence Messages Lead to Terminations of Otherwise Wanted Pregnancies

In the 1980s, U.S. government official recommendations regarding alcohol use in pregnancy shifted from focusing on FAS and high levels of drinking during pregnancy to official recommendations of complete abstinence. Similar shifts—to either abstinence or to no more than low levels of drinking during pregnancy—occurred in other English speaking countries [58]. In relation to these shifted recommendations and associated health messages, new abortion-related concerns emerged. Specifically, a concern emerged that recommending complete abstinence from alcohol use during pregnancy would lead women to terminate otherwise wanted pregnancies [35,59,60,61,62]. This concern also appears in obstetrics and gynecology professional association guidelines, which state that low levels of drinking in early pregnancy are not indications for terminating pregnancies [63,64]. Connecting back to the lack of research about whether there is a threshold of drinking during pregnancy at which providers might recommend consideration of abortion, these guidelines do not state whether there is a level of drinking that is an indication for abortion.

One Australian study has documented the alcohol industry as a source of expressed concerns that health messages about alcohol and pregnancy (such as on warning labels) lead women to terminate pregnancies [65]. The alcohol industry has engaged in persistent efforts to manipulate scientists as well as policymakers to avoid policies that reduce overall alcohol consumption and thus their commercial interests [66,67]. The fact that the alcohol industry appears to be a source of this type of abortion aversion raises questions about the purpose of raising these concerns, and whether these concerns are distractions from other policy approaches that might hurt alcohol industry profits.

This type of abortion aversion is not innocuous. It appears to prioritize avoiding abortion over strategies that might help improve the health of women and fetuses in relation to alcohol. For example, a 2012 paper found no evidence for the argument that recommending abstinence from alcohol during pregnancy would lead women to terminate otherwise wanted pregnancies [68]. Alongside this paper, the journal published a commentary arguing that policymakers (in Australia) could feel comfortable recommending abstinence from alcohol during pregnancy, as they no longer had to worry that this recommendation would lead women to have abortions [69]. What the commentary failed to acknowledge was that there was no evidence or recent change in evidence about whether recommending abstinence actually translated into reduced alcohol use during pregnancy compared to no recommendation, or to a no more than low-level recommendation. More recent research has not found a difference in alcohol consumption during pregnancy between abstinence recommendations versus previous recommendations [70], which is consistent with individual-level counseling message research [71]. Essentially, though, the commentary argued that we should base alcohol/pregnancy policies on whether they lead women to have abortions, rather than whether they effectively reduce alcohol use during pregnancy or related harms. Effective public health policies are not created by only arguing that they do not have unintended effects—rather, they should be evaluated on whether they have intended effects and then whether they also have unintended effects. In this case, abortion aversion affected policy arguments related to alcohol use during pregnancy, and has likely affected the types of policies related to alcohol use during pregnancy that are imagined and thus implemented.

## 4. Presumption of Abortion as Negative Pregnancy Outcome

There is clear evidence that denying women abortions leads to more economic insecurity, worse short term physical health, and to violence from the man involved in the pregnancy, and that abortion is safer than childbirth [72,73,74,75]. Despite this evidence, public health agencies [76] and public health researchers often treat abortion as a negative outcome that should be prevented [77,78].

This same theme of abortion as a negative outcome exists in the literature regarding alcohol (and drug) use during pregnancy. A common way that this appears is within articles examining factors (including or focused explicitly on alcohol and drug use) associated with having an abortion (compared to birth) or multiple abortions (compared to one) [79,80,81,82,83,84]. While often not stated explicitly, the logic is that (heavier levels of) alcohol use among women of reproductive age leads to unintended pregnancy, which then leads to abortion (framed as a negative outcome), and to multiple abortions (framed as even more negative outcomes).

## 5. Implications

The role of abortion aversion in influencing alcohol/pregnancy policies matters from a public health perspective. Despite more than 40 years of policymaking related to alcohol use during pregnancy, alcohol use during pregnancy has remained relatively steady in the U.S. since the 1990s [9,10,11,12,13]. Combined with the evidence that extant alcohol/pregnancy policies do not effectively reduce the harms from alcohol use during pregnancy, and that they may increase these harms [16,17,18], that alcohol use during pregnancy has remained steady indicates that new policy approaches are needed. As a first step in developing policies that can effectively reduce alcohol use and related harms during pregnancy, scientists, health professionals, and public health professionals need to explore how our own explicit or implicit abortion aversion has influenced the types of scientific questions we ask, and the ways in which we frame policy conversations. Only after coming to terms with how abortion aversion has influenced our science and policymaking to date can we move forward in crafting new questions and new possible solutions.

Removing abortion aversion from the science related to alcohol use during pregnancy might allow us to pursue research questions such as: What policies, structures, and services are necessary to ensure that women with alcohol and drug use disorders are able to obtain wanted abortions? What is the rate of adverse outcomes at different levels and patterns of alcohol use during pregnancy when combined with different social and nutritional characteristics? Which, if any, health messages and guidelines around drinking during pregnancy reduce alcohol consumption during pregnancy, as well as the related harms?

## 6. Conclusions

Abortion aversion appears to influence science and thus policymaking related to alcohol use during pregnancy, which is to the detriment of developing and adopting policies that reduce harms from alcohol use during pregnancy. To conceptualize and then develop and enact policies that may be more likely to reduce alcohol use during pregnancy and related harms, scientists, clinicians, and policy influencers need to become aware of how aversion to abortion has influenced science and policymaking to date, and engage in intentional practices to avoid being influenced by abortion aversion moving forward.

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
