# Peer review of "The Presence and Consequences of Abortion Aversion in Scientific Research Related to Alcohol Use during Pregnancy"

_ijerph, 2019, doi:10.3390/ijerph16162888_

Round 1
Reviewer 1 Report
Thank you for the opportunity to review the manuscript. I think the paper is thought-provoking for researchers, clinicians and policy makers. However, I would recommend some major changes. An important consideration is that the author states that it is a “conceptual paper” however, it has been submitted as an original article and the way that the paper is described comes across more as a review paper. However, if the author chose – they could re-write as a conceptual paper – but this journal may not have this type of submission – so could fit better under the “review” category.
Furthermore, a major concern is that this paper is over-simplifying the discussion of risk. Further discussion is required regarding the current research regarding the levels of prenatal alcohol exposure and potential risk. It is vital to discuss that multiple factors are involved i.e. it is not the alcohol exposure alone that determines risk e.g. nutrition, stress, genetics – it is much more complicated than just the alcohol exposure e.g. see May et al 2011 “Maternal risk factors for fetal alcohol spectrum disorders: not as simple as it might seem. Additionally, it is important to discuss that are instances where children are exposed to high levels of prenatal alcohol, but are non-syndromal. These areas are all vital considerations for health professionals in providing recommendations.
Title: would be good for the title to reflect the type of paper this up front so that readers know from the outset e.g. review
Abstract
Line 8 – (and throughout) – References to U.S policies – state and/or federal? – throughout the paper would be helpful to provide more clarification about differences between different states and compared to federal policy for readers. E.g. in intro how many/what states do have more punitive policies?
Introduction
Would suggest that there needs to be a clear definition of what is meant by the term “abortion aversion”
Would be good to include a clear description of the aim of the current study at the end of the Introduction section.
Page 2 – line 15 – stated that policies have become more punitive – would be good to provide examples and explain what this means – what do these punitive polices included?
Line 34 – 36 – too small to be a paragraph and same at line 27 – 29
Materials and Methods
Line 31 – as discussed in the overall comments at the start – states that is a “conceptual paper” – however this is not consistent with the way the paper is presented - need to decide if going to present as a conceptual paper or as a review and re-write accordingly.
Line 37 – it is stated that a thematic analysis was undertaken – and Braun and Clark is referenced –How was coding undertaken? How were themes generated? What aspects of Braun and Clark were followed? Much more details are required to strengthen this section if a thematic analysis was undertaken.
Line 38 and as suggested for the Introduction - definition of abortion aversion - again would be good to clearly describe here what is considered as abortion averse. Even though this is not a systematic review further information is required regarding how papers were identified as being abortion averse.
Again – although this is not a systematic review - more information should to provided regarding how papers were selected– e.g. line 35 “Potentially relevant abstracts” – need to describe what this means
Line 43 – 45 – “a key component of this process was noting the often unstated assumptions that underlie the scientific questions” – the way that this is currently re-written could be misinterpreted – I would suggest referring to Braun and Clark and other relevant references regarding interpretative analysis
Line 47 – “To contextualise findings the author then brought in other literature…” would delete this
Page 6 – line 8 – section presumption of abortion as negative pregnancy outcome – notably less discussion of this theme compared to the other themes. Have only considered the physical health outcomes
Line 16 – “while often not stated explicitly” - same as above
Results
Information throughout this section that would be better in the Discussion section – if this was a thematic analysis – would be better to present the results of the analysis more clearly including more selections of extracts to illustrate/support the analysis – then in the Discussion you could bring in other relevant research to support particular points.
Discussion
A limited discussion is currently provided – more examination of potential implications across multiple levels is needed e.g. women, families, service-providers, organisations and at the policy level. What about – particularly in the U.S the legal ramifications for health professionals? – at what level of alcohol exposure are you certain as a health professional you should be recommending abortion? I would suggest health professionals are more broadly risk averse due to current contextual issues. More discussion of relevant contextual issues is required – including broader examination of other potentially influencing factors.
Also vitally important as an outcome of this paper is directions for future research i.e. I would be very interested to know what are women’s experiences and opinions regarding this – what information do women what to be provided if they have consumed high levels of alcohol? Do they want to be presented with abortion as a potential option and to be able to discuss this with their healthcare provider? Has this been examined in the research at all previously? Important area to discuss for future research to consider. Also what about from the health providers perspectives? What are their experiences and needs regarding this area? Many suggestions could be made to guide future research.
Reviewer 2 Report
The main concern is what is the focus of the article. What are the contributions the authors are making in the literature? Currently it seems that it is an opinion-based article using citations from the literature to support their arguments. However, the conclusions drawn from these citations are often misleading (at best). Several times the authors interpret the lack of discussion on abortion as evidence of abortion-aversion. This is unjustified as the findings can reflect a myriad of other things and not abortion aversion as the authors claim throughout the paper. For example, the authors claim that “Importantly, abortion aversion played a role in contributing to this (seemingly counterproductive) effort.” But, how do the authors reach that conclusion? Did the study control explicitly for abortion aversion? It did not. Similarly, data limitations are not the same as “failure to acknowledge” as the authors state towards the end of the second paragraph on page 4. If information on abortions was not available in the dataset, the cited study could not use abortion as an outcome. It does not mean that they deliberately excluded abortion because they had a preference against abortion. Another example of randomly attributing research findings to abortion aversion is on page 5 when discussing the Australian alcohol industry. Why would the authors conclude that the health messages are “at least in part abortion aversion”? As a health industry your goal is to maximize profits. Therefore, any potential attempts to influence overall alcohol consumption is not due to abortion aversion, but profit maximizing behavior of the firms. The term “aversion” has to do with preferences of individuals and their respective utility. However, firms operate under a different goal: they are profit and not utility maximizers. The same holds when talking about the policy-makers on page 6. Policy-makers make decisions based on a cost-benefit analysis and as long as the benefits outweigh the costs they introduce a policy. One can argue that abortion aversion comes into the cost-benefit analysis, however, adoption a policy is not automatically due to abortion aversion. In general, anything we do not know cannot be immediately attributed to abortion aversion.
This problem is evident as early as in the Introduction where the authors give the impression that low birthweight and preterm births are due to alcohol/pregnancy policies. However, this is not accurate. The studies cited do not examine any causal effects, thus, this argument is misleading. This is true throughout the paper. There is no discussion about the distinction between correlation and causal studies. As a matter of fact, none of the cited studies appear to identify causality. Therefore, the authors cannot argue that anything that the studies do not directly argue about is due to abortion aversion. It could be due to a myriad of other unobserved traits. If such traits are not controlled for, arguing that the research is due to abortion aversion is completely a personal view without any scientific justification. This is reflected also in the Methods section where the authors give vague descriptions about how they chose the papers. There is no information about what the criteria were to retain relevant abstracts (and why only abstracts and not looking the whole paper!) and how they decided that abortion aversion was “implicitly or explicitly framed as negative in the paper.”
If this paper is about examining why the research does not look into abortion as an outcome, it should be much more comprehensive and objective. Even if we believe that it is about abortion aversion (which we cannot unless the authors provide evidence using actual data analysis), why would we compare studies from vastly different countries? That’s what the authors currently do on page 6 under 3.3, when they treat Australia and China as two countries that are similar. If it is true that there are differences towards abortion, such preferences might be stronger in China and less strong in Australia due to cultural reasons and one child policies.
Finally, under 3.1.1., the citations are old and do not necessarily reflect the what is happening today, and there are several sentences (even paragraphs) that are highly unclear throughout the paper:
· Last paragraph on p.3.
· Last sentence, first paragraph on p.4.
· First sentence, second paragraph on p.4.
· Last two sentences, last paragraph on p.4. In that same paragraph,
Overall, the limitations stated in the last paragraph on page 6 are severe. If the authors want to discuss abortion aversion they need to follow a more robust scientific methodology. The second limitation itself is actually arguing that the paper in its current state has no merit since it cannot address any of these problems, and as such it is no different than an opinion-based article.
Reviewer 3 Report
this is a structured commentary highlighting how assumptions about abortion relate to the framing of scientific research findings regarding alcohol use in pregnancy (as well as public health messaging surrounding alcohol use in pregnancy)
it is overall well reasoned, well written, very interesting and likely impactful
my main comment relates to the methods: a single author with deep knowledge in the subject area utilized thematic analysis to identify 3 themes of abortion aversion - these themes are then detailed through both the literature from which they were identified as well as contextualized in other literature with which the author was familiar.
1) i am a touch unclear which parts are which in the results (the literature searched for themes and that wherein the theme is contextualized) - i am not sure that this matters in terms of the results and conclusions - i just didn't see how the methods matched the results.
2) the author is correct in the limitations section: this is not a systematic review (although i am unclear how a systematic review of a thematic analysis would work). a perhaps more important potential limitation is that of a single author. as a reader i find the organization of the abortion aversion themes to resonate with my knowledge and experience. however one of the advantages of a systematic review process for an analysis such as this one would be the inclusion of other perspectives - the iterative process of abstract and full text review between >1 person yielding an organizational structure that is more transparent and reproducible. people are acknowledged so presumably this was done?
some smaller things:
1) the discussion of the "botched" CDC statement(s) on alcohol and pregnancy and contraception are excellent - however the content of the original CDC messaging may not be familiar to all readers, and a bit more detail of what was originally stated by CDC may be helpful
2) there is one instance of the use of the term "abuse" - recommend using either use, misuse or use disorder in place of that
3) the writing could benefit from some tightening - the first 2 sentences of the abstract for example - as well as in other places where terms such as "substance use in pregnancy" are repeated in clauses and sentences
Round 2
Reviewer 1 Report
I think this paper works much better as a commentary. However, it still presents as quite a one-sided piece without consideration for the difficulties of health professionals regarding in what circumstances they would recommend abortion - given the complex interplay of prenatal alcohol exposure and other risk factors.
Final section on presumption of abortion as a negative outcome is very small compared to the other sections - currently looks like an after thought rather than a developed section.
I still feel there is a limited discussion provided. Given the wide range of issues that this commentary raises it would be good to provide more discussion to guide future research.
Author Response
I think this paper works much better as a commentary.Thank you for this feedback
However, it still presents as quite a one-sided piece without consideration for the difficulties of health professionals regarding in what circumstances they would recommend abortion - given the complex interplay of prenatal alcohol exposure and other risk factors.I agree that figuring out whether and when health professionals might counsel someone to consider abortion in relation to drinking during pregnancy is important. My argument is that (part of) why we do not have answers to the question you pose is that aversion to abortion has gotten in the way of people prioritizing this science. In response to comment # 4, I also added a recommendation for further research into this area.
Final section on presumption of abortion as a negative outcome is very small compared to the other sections - currently looks like an after thought rather than a developed section.I agree that it is shorter. It is just a much more straightforward section. There isn’t much nuance to it at all. It does not make sense to me to add words for the sake of making it seem of similar importance.
I still feel there is a limited discussion provided. Given the wide range of issues that this commentary raises it would be good to provide more discussion to guide future research.Thank you for pushing me to articulate what some of these research questions might be. I have now added a paragraph to the discussion (lines 251-257) that names three research questions that I think might be pursued if our science related to alcohol use during pregnancy was not constrained by abortion aversion.